# Deciphering the Transcriptomic Complexity of Yak Skin Across Different Ages and Body Sites

**DOI:** 10.3390/ijms26104601

**Published:** 2025-05-11

**Authors:** Xiaolan Zhang, Bingang Shi, Zhidong Zhao, Yunqi Deng, Xuelan Zhou, Jiang Hu

**Affiliations:** College of Animal Science and Technology, Gansu Agricultural University, Lanzhou 730070, China; zhangxiaolan@gsau.edu.cn (X.Z.); shibg@gsau.edu.cn (B.S.); zhaozd@gsau.edu.cn (Z.Z.); 15002555696@163.com (Y.D.); 14201@gsau.edu.cn (X.Z.)

**Keywords:** skin transcriptome, yak, ages, WGCNA

## Abstract

Differences in skin and hair phenotypes between the scapular and ventral regions of yaks (Bos grunniens) are obvious and become more prominent with age. However, the genetic mechanism that causes differences in yak skin at different ages has not been reported. In this study, we investigated the transcriptomic profile of yak skin across different ages (0.5 years, 2.5 years, and 4.5 years) and body sites (scapular and ventral regions). Differential gene expression analysis was initially conducted to explore the transcriptomic differences in skin at different ages and different body sites. Subsequently, weighted gene co-expression network analysis (WGCNA) was employed to analyze the transcriptomic data comprehensively. The results showed that, among all comparison groups, the Y2.5_S vs. Y2.5_V group (regional comparison) exhibited the highest number of DEGs, with 491 genes (179 upregulated and 312 downregulated), followed by the Y2.5_V vs. Y0.5_V group (age comparison), which had 370 DEGs (103 upregulated and 267 downregulated). DEGs such as *IGF2BP3*, *ADCY8*, *FOSL1*, and *S100A8* were found in all comparison groups of different ages, and multiple members of the HOX gene family including *HOXC10*, *HOXA9*, *HOXA6*, *HOXB9*, and *HOXB6* were differentially expressed in comparison groups at different sites. Functional enrichment analysis showed that there were more obvious differences in immune function between different ages of skin and more obvious differences in endocrine function between different parts of skin. WGCNA revealed that genes related with immunity such as *GLYATL2*, *ACSL5*, and *SPDEF* were the core genes of the co-expression module associated with the scapula region, and multiple genes related to hair follicle development such as *FOXN1*, *OVOL1*, *DLX3*, *HOXC13*, and *TCHH* were found to be the hub genes of the co-expression module associated with the ventral region. Overall, our study provides valuable insights into the transcriptomic complexity of yak skin across different ages and body sites. The differential gene expression patterns and co-expression network modules identified in this study lay the foundation for further research on skin biology and adaptation mechanisms in yaks.

## 1. Introduction

The yak (Bos grunniens) is a vital livestock native to the high-altitude regions of the Tibetan Plateau. Yaks are well-adapted to extreme environmental conditions, including low oxygen levels, cold temperatures, and high UV radiation exposure, and they play a crucial role in the local economy and ecosystem [1]. As the first line of defense against external environmental stress, yak skin not only plays an important role in resisting harsh environmental conditions and maintaining homeostasis but also plays an indispensable function in immune defense and sensory perception. Understanding yak biology, especially the intricacies of their skin tissues, is crucial for conservation and sustainable utilization strategies [2,3].

The skin is primarily composed of two layers: epidermis and dermis. The epidermis, which consists of predominantly keratinocytes, plays major roles in the maintenance of homeostasis, protecting the organism from physical, chemical, and infectious insults. Also, the epidermis contains UV-resistant melanocytes and Langerhans cells, mediating the immune recognition of foreign pathogens [4,5]. The underlying dermis is filled with a complex extracellular matrix (ECM) and dermal fibroblasts that provide the skin with strength and resilience and is crucial for the structural maintenance of skin integrity [6,7]. Like all organ systems, the skin also undergoes multiple changes with aging, and these changes are relatively obvious and vulnerable to the external environment. In humans, the hallmark features of skin aging involve structural and functional changes, including decreased collagen content, reduced skin thickness, the formation of wrinkles, and changes in immune function [8]. Since 1983, the skin has been recognized as a primary immunologic organ, which is involved in constant immune surveillance to intercept and destroy harmful substances and antigens in direct contact with the external environment [9]. Many studies have revealed a close relationship between skin aging and disturbances in the skin immune system [8,10].

The study found that in yaks, the thickness of yak skin, epidermis, and dermis increased from newborn to adult, and skin thickness decreased from dorsally to ventrally on the trunk [11]. Up to now, studies on the molecular aspects of skin aging mainly focus on humans and mice, and there are few studies on yaks [6,12,13] Radiation is one of the main external enemies of cutaneous juvenescence, and it is even more severe for yaks because they live in an environment highly exposed to ultraviolet (UV) light. It was observed that the back hair of older yaks was obviously more prone to shedding, which may be related to the changes in skin immune and anti-inflammatory functions as well as long-term intense UV exposure to the back skin. However, research on many aspects of yak skin biology at the molecular level remains unexplored.

Long ventral hair is one of the most typical phenotypic characteristics of yaks. The skin in different parts of yaks presented a different hair feature, such as more villi (produced by secondary hair follicles) in the scapula region, and obvious seasonal shedding with the development of hair follicles, while the ventral hair was longer and more coarse hair (produced by primary hair follicles), without obvious seasonal shedding. This provides a good model for studying the difference between secondary and primary hair follicle development. In addition, compared with the back skin, the ventral skin may be less exposed to UV light. Combined with these differences from the phenotype and the external environment, the scapular and ventral skin of yaks may produce more differences in gene expression during their growth.

Although chronological and environmentally induced components are major factors in skin aging, there may also be significant differences in some individuals who appear to be relatively resistant to eventual skin aging [14]; that is, genetic differences may play an important role. A genome-wide association study found that SNPs associated with the MCIR gene may play a role in skin aging [15]. Studies on mammalian skin transcriptomics primarily focus on the research of hair follicle cycle development in wool-producing animals including yaks or comparative studies on wool fineness [16,17,18,19]. In terms of skin adaptability, Wu et al. revealed the drivers of plateau adaptability in cashmere goats through genomic and skin transcriptomic analyses of Tibetan sheep and Jiangnan sheep [20]. The unique characteristics of yak skin may provide an excellent model for studying the biological functions and immune properties of the skin.

In this study, we hypothesize that yak skin has age - and region-specific gene expression patterns, which may be linked to the development and adaptation of yak skin. Given the large volume and complexity of transcriptomic data from different ages and parts of yak skin, WGCNA was applied to classify genes with similar expression patterns, facilitating a deeper understanding of the intricate interactions and coordinated regulatory network among genes. Our research on the transcriptomic profiles and functional studies of yak skin across different ages and body sites would provide insights into the molecular mechanisms underlying skin biology and adaptation mechanisms and offers a theoretical basis for the genetic breeding and resource protection of yaks.

## 2. Results

### 2.1. Overview of RNA-seq

To study the transcriptome differences in yak skin tissues at different ages and different parts, RNA-seq sequencing was performed on the skin from the scapular region at different ages, including 0.5 years old (Y0.5_S), 2.5 years old (Y2.5_S), and 4.5 years old (Y4.5_S), as well as from the ventral region at 0.5 years old (Y0.5_V) and 2.5 years old (Y2.5_V). After quality control, high-quality clean reads were obtained, with an effective data volume of 6.86-7.05 G per sample, and the Q30 values ranged from 95.74% to 96.45%, with an average GC content of 50.52%. The mapping rate to the yak reference genome using HISAT2 ranged from 93.27% to 96.5% (Appendix A), indicating reliable sequencing data suitable for subsequent analysis. The distribution of gene expression based on the FPKM value across the thirty samples were statistically analyzed, as represented by a stacked bar chart (Figure 1A). The PCA results indicated that the same groups are distinctly clustered together, while the Y2.5_S group was not clearly separated from Y4.5_S (Figure 1B), suggesting that the genetic and physiological changes in skin and hair tended to weaken as the development of yak skin and hair matured with age.

Among all the comparison groups, the largest number of DEGs was observed between the Y2.5_S vs. Y2.5_V group, with a total count of 491 (179upregulated and 312downregulated). A total of 370 DEGs (103upregulated and 267downregulated) were screened in the Y2.5_V vs. Y0.5_V group, which was also relatively large. Only 19 DEGs (10 upregulated and 9 downregulated) were found in the Y4.5_S vs. Y2.5_S group (Figure 2A). Correspondingly, there were a relatively large number of independent DEGs in the comparison groups of Y2.5_S vs. Y2.5_V (259) and Y2.5_V vs. Y0.5_V (210). Notably, in the two comparison groups of different regions, Y0.5_S vs. Y0.5_V and Y2.5_S vs. Y2.5_V, there were 118 common DEGs, while in the comparison groups of different ages, Y2.5_S vs. Y0.5_S and Y2.5_V vs. Y0.5_V, only 46 common DEGs were found (Figure 2B).

### 2.2. Expression Patterns and Functional Analysis of DEGs at Different Ages

To explore the crucial genes influencing the physiological changes in yak skin during different growth stages, the expression patterns of DEGs at different ages were analyzed separately. The cluster heat maps were used to analyze the DEGs in the scapular and the DEGs in ventral regions during different ages, respectively. The result of the clustering analysis indicated that the gene expression patterns of yak skin in 0.5 years old were obviously different from other ages, both in the scapula and the ventral region. The gene expression pattern of the skin at 2.5 years old and 4.5 years old was similar in the scapular region and was not completely separated (Figure 3 A,B). Then, GO and KEGG enrichment analyses were performed on DEGs across comparative groups of different ages (Y2.5_S vs. Y0.5_S, Y4.5_S vs. Y0.5_S, and Y2.5_V vs. Y0.5_V). KEGG pathway analysis revealed that the pathways related to immunity and inflammation, such as the IL-17 signaling pathway and the cytokine–cytokine receptor interaction signaling pathway, were significantly enriched in all three comparison groups; the TNF signaling pathway and chemokine signaling pathway were significantly enriched both in the comparison groups of Y2.5_S vs. Y0.5_S and Y2.5_V vs. Y0.5_V. Furthermore, the pathways related to drug metabolism, such as drug metabolism–other enzymes, and the metabolism of xenobiotics by cytochrome P450 signaling pathways were specifically enriched when comparing the Y2.5_S vs. Y0.5_S group. The pathways related to infection including staphylococcus aureus infection and pertussis were specifically enriched in the contrast of the Y4.5_S vs. Y0.5_S group, and the pathways related to hormones including thyroid hormone synthesis and the estrogen signaling pathway were specifically enriched when comparing the Y2.5_V vs. Y0.5_V group (Figure 3C). For GO analysis, in the biological process (BP) category, the GO terms related to immunity such as the innate immune response, the immune response, the chemokine-mediated signaling pathway, and natural killer cell-mediated cytotoxicity were enriched into at least one comparison group. The GO terms related to collagen including the collagen catabolic process, collagen fibril organization, and the collagen biosynthetic process were significantly enriched in the Y2.5_S vs. Y0.5_S or Y2.5_V vs. Y0.5_V contrast group, while no difference was observed in the Y4.5_S vs. Y0.5_S group. In the cellular component (CC) category, GO terms such as the extracellular matrix, the extracellular space, and the extracellular region were primarily enriched for the DEGs in all three comparison groups. In the molecular function (MF) category, chemokine activity, peptidoglycan binding, protease binding, and serine-type endopeptidase inhibitor activity were enriched within at least two comparison groups, suggesting that these GO terms may be related to molecular changes in yak skin with age (Figure 3D). Finally, 23 genes that were differentially expressed in all the three comparison groups of different ages were presented using a cluster heat map, and genes of the same age could be found clustered together (Figure 2B and Figure 3E).

### 2.3. Functional Analysis of DEGs at Different Skin Regions

Subsequently, the DEGs in different regions of the yak skin were analyzed separately. The volcano map was used to show the DEGs of the two comparison groups in different regions, and it was obvious that the number of DEGs of the scapular and ventral parts at 2.5 years old was far more than that at 0.5 years old. In addition, among the top ten DEGs, multiple DEGs such as *HOXA9*, *HOXC10*, *CALCR*, *TYRP1*, and *DMRTA1* were found in the two comparison groups (Figure 4A). KEGG enrichment analysis of the two groups showed that more than half of the top 20 enriched signaling pathways were present in both comparison groups (Figure 4B). The co-enriched signaling pathway was mainly involved in staphylococcus aureus infection, vascular smooth muscle contraction, adrenergic signaling in cardiomyocytes, and neuroactive ligand–receptor interaction. In addition, the IL-17 signaling pathway and cytokine–cytokine receptor interaction related to immunity were specifically enriched in comparison group Y2.5_Svs. Y2.5_V. Similarly, multiple identical GO terms were also primarily enriched in the two comparison groups, such as keratinization, embryonic skeletal system morphogenesis, keratin filament, intermediate filament, and structural molecule activity, which mainly related to the structure composition and development of skin and hair. Notably, hair follicle morphogenesis and peptide cross-linking were enriched in Y2.5_Svs. Y2.5_V, which may be related to the larger differences in follicle development and hair characteristics between the scapular and ventral regions at 2.5 years old (Figure 4C). Since these two groups had the largest number of common DEGs (Figure 2B), PPI analysis of the shared DEGs showed that multiple members of the HOX family and TAT were located in the core of the PPI network (Figure 4D).

### 2.4. Weighted Gene Co-Expression Network Analysis

To explore the genes and biological functions that could comprehensively reflect different ages and different regions in yak skin, 2355 genes screened with a threshold of P adjust < 0.05 and FC ≥ 1.5 in all comparison groups were used for WGCNA. Eight co-expression modules were established (Figure 5A). The number of genes in different modules varied widely, from 40 genes in the pink module to 978 in the turquoise module (Appendix A). The heat map of the module–trait relationship showed that the pink module was significantly correlated with the scapula group (cor = 0.62, *p* = 3 × 10^−4^), the turquoise module was strongly correlated with the ventral group (cor = 0.85, *p* = 3 × 10^−9^), the red module was significantly correlated with age 2.5 years (cor = 0.67, *p* = 5 × 10^−5^), and the blue module was strongly correlated with age 0.5 years (cor = 0.84, *p* = 7 × 10^−9^) (Figure 5B). Therefore, the genes within these four modules were selected for further KEGG analysis. KEGG enrichment results showed that the pathways involved in hair development such as melanogenesis, the hedgehog signaling pathway, and the wnt signaling pathway were enriched in the blue module. The pathways such as mineral absorption, tyrosine metabolism, ferroptosis, vascular smooth muscle contraction, cardiac muscle contraction, and neuroactive ligand–receptor interaction, which were enriched in the modules (pink or turquoise) related to body site, also appeared in the KEGG results of the DEGs for different skin regions (Figure 4B and Figure 5C). Similarly, the pathways such as the cAMP signaling pathway, ECM–receptor interaction, GABAergic synapse, amoebiasis, and protein digestion and absorption were specifically enriched in the modules (blue or red) that were related to ages and also appeared in the KEGG enrichment results of the DEGs for different ages (Figure 3C and Figure 5C). To further investigate the key genes and their co-expression networks in specific modules, the hub genes of the four module networks were calculated and visualized using the Cytoscape software. The result showed that *GLYATL2* and *ACSL5* were the hub genes in the pink module. *FOXN1*, *KRT27*, and *OVOL1* were the core genes in the turquoise network, and multiple genes related to hair follicle development such as *HOXC13*, *TCHH*, *DLX3*, and *BAMBI* were present in the turquoise network. *HOXD4*, *GPC3*, and *NAV3* were found to be the hub genes for the blue module, and *S100A2*, *KRT6A*, *TGFA*, and *KLK7* were the hub genes in the red module (Figure 6).

### 2.5. RT-qPCR Validation of DEGs

To verify RNA-seq results, six DEGs *(DUSP14*, *FOXA1*, *GPC3*, *MC5R*, *SPDEF*, and *FOXN1*) were randomly selected for RT-qPCR validation. The trend of the RT-qPCR results of the six genes was consistent with that of the gene expression pattern of RNA-seq (Figure 7), indicating that the transcriptome sequencing results were reliable.

## 3. Discussion

Skin aging is considered to be related to chronologic, radiation, immunity, infection, microbiome, and diseases in humans, and studies have shown that the skin robustly mirrors aging in more internal tissues [21]. From this perspective, skin phenotypes can also serve as a reference indicator in livestock breeding selection and management. Yak skin and coat are of great significance due to the yaks’ long-term adaptation to the alpine environment. The age of 0.5 years is the juvenile stage of yaks, while an age of 2.5 years, which marks the onset of sexual maturity, corresponds to the young adult stage, and 4.5 years old is regarded as the adult stage. These three age groups correspond to different growth and development stages of yaks, and their physiological characteristics and growth conditions are significantly different.

In this study, the transcriptomic profile of yak skin across different ages and two body sites was investigated. Principal component and cluster analysis of all samples showed that 2.5-year-old and 4.5-year-old yaks were not distinguishable in terms of skin gene expression, but both were clearly separated from 0.5-year-old yaks, implying that the skin and coat of 0.5-year-old yaks were not fully developed, and gene expression changes slowed down after 2.5 years old. Among all comparison groups, the Y2.5_S-vs.-Y2.5_V group had the most obvious difference and the highest number of DEGs, suggesting that the changes in gene expression in different regions (scapula and ventral regions) of yak skin after birth were greater than those in the same parts at different ages. The dermis of different body regions has different embryonic origins. The dorsal dermis is derived from the paraxial mesoderm, and the abdominal dermis develops from the lateral plate mesoderm [22]. This serves as the basis for the functional and anatomical differences in the skin in different body regions. For yaks, the gene expression difference between the scapular and ventral skin increased gradually with age, which may be due to the gradual perfection of differentiation of primary and secondary hair follicles and the difference in external environment such as sunlight in the growth of yaks.

The KEGG enrichment analysis of the DEGs in skin tissues of different ages showed that multiple immune-related signaling pathways, such as the TNF signaling pathway, the IL-17 signaling pathway, and cytokine–cytokine receptor interaction, were significantly enriched in the age-difference comparison groups. Furthermore, several endocrine and nervous system-related signaling pathways, such as thyroid hormone synthesis, the estrogen signaling pathway, and neuroactive ligand–receptor interaction, were specifically and significantly enriched in the Y2.5_V-vs.-Y0.5_V group. The skin is a crucial part of the immune system, which undergoes aging with the extension of life spans [23]. It was reported that blocking the activity of IL-17 signaling is considered a potential strategy to delay skin aging [24]. In addition, skin is considered a key neuron–endocrine system, and it playing an important role in the regulation of immune cells in skin [25,26]. Signaling pathways associated with infection and autoimmune diseases, such as staphylococcus aureus infection, pertussis, and systemic lupus erythematosus, were specifically enriched in the Y4.5_S-vs.-Y0.5_S group. This may be linked to the fact that although yaks at the age of 4.5 years old are in their middle-age stage, the skin phenotype of these yaks, especially that of the dorsal skin, shows the characteristics of being prone to hair loss and pigment deposition. For GO enrichment analysis, several terms related to the extracellular matrix and collagen (the main components of dermis) were enriched in different age-related comparison groups, indicating that the dermis structure of yaks changed with age. The study found that over time, old dermal fibroblasts lose their identity, manifested by decreased expression of genes involved in the formation of the extracellular matrix [27]. The cornified envelope was significantly enriched in the Y2.5_S-vs.-Y0.5_S and Y4.5_S-vs.-Y0.5_S groups. However, it was not significantly enriched in the Y2.5_V-vs.-Y0.5_V comparison group. This may be related to more UV exposure to the scapular region compared to the ventral region, given that the cornified layer acts as the outermost interface between the underlying epidermal and dermal tissues and the atmosphere [10].

For the analysis of DEGs in different regions, the most obvious is that multiple members of the HOX family, such as *HOXA6*, *HOXA9*, *HOXC12*, and *HOXC10*, are up-regulated in the ventral skin. HOX genes exhibit spatial and temporal changes in expression during human skin development and have different functions in skin development [28,29]. In mice, it was reported that compared with the skin tissues of the back and abdomen, the expression profiles of multiple members of the HOX family are lower, or even absent, in the cheek [22]. The DEGs related to melanin deposition or immunity, including *TYR*, *TYRP1* [30], *MC5R* [31], *CXCL13* [32], *IL17C*, and *IL17F* [33], are expressed at a higher level in the scapular region as they enhance the skin’s resistance to ultraviolet radiation. This may be associated with the fact that the skin on the back of the yak is exposed to more ultraviolet radiation, which is also a manifestation of the adaptability of yak skin tissue to the external environment. This finding is consistent with the increased expression of multiple interleukins and chemokines (ILs and CXCLs) in the dorsal skin of humans and mice following ultraviolet radiation induction [34].

Among these significantly enriched pathways that were co-enriched in the two comparison groups at different regions, several endocrine-related pathways such as the estrogen signaling pathway, adrenergic signaling in cardiomyocytes, and neuroactive ligand–receptor interaction are included in particular. Compared with the above enrichment results between different ages and combined with KEGG secondary classification enrichment analysis (Appendix A), it can be concluded that the skin of different ages has more significant differences in the immune system, while the skin of different regions with more significant differences in the endocrine system may be related to the adaptability of the yak skin to the external environment and physiological functions. GO terms that are related to keratinization and skeletal system development were mainly enriched in the DEGs of the two comparison groups of different regions. Keratin is the main component of synthetic skin epidermis and hair [35], and hair follicle morphogenesis is enriched in the Y2.5_S-vs.-Y2.5_V group, suggesting that keratin is related to the differences in hair follicle morphogenesis and epidermis formation in the skin of the two regions. Studies have found that there is a link between skin development and the skeletal system, but the relevant mechanism is not well understood [36].

Four gene co-expression networks were constructed using WGCNA. Genes *GLYATL2*, *ACSL5*, *SPDEF*, and *FOXA1* were found to be key nodes in the pink module associated with the scapula. *GLYATL2* has been reported to play a potential role in barrier function and the immune response in humans [37]. *ACSL5* was found to function as an immune-dependent tumor suppressor [38], and *SPDEF* and *FOXA1* are the canonical markers of mammalian “goblet cells” [39,40]. It can be further suggested that the scapular skin of yaks plays a stronger role in immune function. Multiple genes associated with hair follicle development and keratin such as *FOXN1*, *OVOL1*, *KRT27*, *KRT71*, *KRT82*, *KRT35*, *KRT39*, *DLX3*, *HOXC13*, and *TCHH* [41,42,43] were found to be the hub genes in the turquoise module. This may be related to the fact that there are more primary hair follicles in the ventral region of yaks, and it is consistent with the study that multiple *KRT* or *KAP* genes in the skin tissue of Jiangnan cashmere goats were significantly higher than those of Tibetan cashmere goats, while Tibetan cashmere goats have finer cashmere than Jiangnan cashmere goats [20]. In the red module significantly associated with 2.5 years old, hub genes such as *S100A2*, *KRT6A*, *TGFA*, and *KLK7* [44,45,46,47] were reported to play roles in skin damage repair, inflammation, and the immune response, which may be related to the enhancement of the barrier and immune function of the yak skin at 2.5 years old. The expression patterns of genes in different anatomical regions of the skin may vary among species. For example, the *SFRP1* gene is upregulated in the ventral skin of yaks, whereas it is more highly expressed in the back skin of pigs. The *COLLA1* gene is more highly expressed in the back skin of pigs [48] but shows no significant difference in expression between the ventral and scapular skin of yaks, while its expression is higher in 0.5-year-old yaks. This may be related to the environmental adaptability of the species during evolution.

In summary, this study aimed to screen specific genes in yaks of different ages from the scapular and ventral regions and analyze their functions to explore the plateau adaptability of yak skin tissues. However, the samples used in this study only included healthy individuals, and the lack of comparative analysis with other plateau species made it difficult to clarify the unique aspects of yak skin adaptability. In addition, the functions and mechanisms of critical specific genes remain to be further verified. Future studies can expand the sample scope and adopt methods such as multi-omics integration and validation through molecular and cellular experiments to further investigate the adaptation mechanisms.

## 4. Materials and Methods

### 4.1. Experimental Animals and Sample Collection

The yaks used in this study were obtained from farmers’ households in Tianzhu County, Gansu Province, China. A total of 6 healthy male yaks (*n* = 6) reared on natural pasture were randomly selected from each of the three age groups (0.5 years old, 2.5 years old, and 4.5 years old), with a total of 18 yaks used for skin sample collection. All yaks fasted for 24 h before slaughter. After slaughter, skin samples were taken from the scapular (*n* = 6) and ventral regions (*n* = 6) of 0.5- and 2.5-year-old yaks and only from the scapular region (*n* = 6) of 4.5-year-old yaks. Thirty skin samples were collected and stored in liquid nitrogen. Before collection, the hair on the skin of the collection sites were cleaned with scissors and disinfected with 75% alcohol to prevent any potential contamination that could affect the RNA quality and subsequent sequencing results.

### 4.2. RNA Extraction, Library Preparation, and Sequencing

Total RNA was extracted from skin samples using the TRIzol reagent (Invitrogen Corporation, Carlsbad, CA, USA) following the manufacturer’s instructions. RNA purity and quantification were evaluated using the NanoDrop 2000 spectrophotometer (Thermo Fisher Scientific, Waltham, MA, USA). RNA integrity was assessed using the Agilent 2100 bioanalyzer (Agilent Technologies, Santa Clara, CA, USA). RNA integrity numbers (RINs) ≥ 6.5 and 28S/18S ratios ≥ 1.0 were used to construct the sequencing library. Then, the libraries were constructed using the VAHTS Universal V6 RNA-seq Library Prep Kit according to the manufacturer’s instructions. The libraries were sequenced on an llumina Novaseq 6000 platform using 150 bp paired-end reads. Raw reads in fastq format were firstly processed using FastQC (Version 0.11.9) software, and low-quality reads were removed to obtain clean reads. The clean reads were mapped to the reference genome (https://www.ncbi.nlm.nih.gov/assembly/GCF_000298355.1, accessed on 29 October 2024) using HISAT2. The FPKM of each gene was calculated, and the read counts of each gene were obtained using HTSeq-count. PCA analysis was performed using R (v 3.2.0) to evaluate the biological duplication of samples. Transcriptome sequencing was conducted by OE Biotech Co., Ltd. (Shanghai, China).

### 4.3. Differentially Expressed Genes and Functional Enrichment Analysis

Differential expression analysis was performed using DESeq2. The false discovery rate (FDR) was obtain by adjusting the *p*-value. FDR < 0.05 and |log2(FoldChange)| ≥ 1 were set as the threshold for significantly differential expression genes (DEGs). Hierarchical cluster analysis of DEGs was performed using R (v 3.2.0) to demonstrate the expression pattern of genes in different groups and samples. The GO (Gene Ontology) and KEGG (Kyoto Encyclopedia of Genes and Genomes) databases were primarily used for gene function and biological pathway enrichment analysis. Based on the hypergeometric distribution, GO and KEGG pathway enrichment analysis of DEGs was performed to screen the significantly enriched terms using R (v 3.2.0). The PPI network was analyzed using the STRING database and visualized using the Cytoscape (V 3.10.2) software.

### 4.4. WGCNA

The WGCNA package was used to construct the gene co-expression network for the integrated analysis of gene expression patterns and regulatory networks in yak skin of different ages and different regions. The genes used to construct the co-expression network were obtained based on FPKM data with a threshold of FC ≥ 1.5 and q value < 0.05. A soft threshold of β = 8 was chosen as it achieved a scale-free topology model (R^2^ > 0.8) (Appendix A), meeting network criteria. Then, the dynamic shear tree algorithm (parameters: minimum module size = 30 genes; similar modules merged when cuttree height < 0.25) was used to identify co-expression modules. The correlations between each module, age, and body sites were calculated, and the modules significantly related to age and sites were screened out. Subsequently, further KEGG analysis was carried out on the genes in the key modules and gene regulatory network data derived from WGCNA. The data were imported into Cytoscape (version 3.10.0) for the visual representation and mining of hub genes using Cytohubba.

### 4.5. RT-qPCR

Total RNA obtained from yak skin tissue samples on all groups was used for RT-qPCR analysis. Six DEGs were randomly selected for verification. cDNA synthesis was carried out using the PrimeScript™ RT kit (Takara, Beijing, China) with the extracted total RNA. The reaction mixture comprised a total volume of 20.0 μL, consisting of 10.0 μL of the SYBR qPCR Master Mix (Takara, Beijing, China), 0.5 μL of each of the positive and negative primers, 200 ng of the cDNA template, and ddH2O as a supplement. The amplification procedure included a pre-denaturation step at 95 °C for 30 s, denaturation at 95 °C for 10 s, annealing at 60 °C for 30 s, and extension at 65 °C for 5 s, repeated for 40 cycles. GAPDH served as the internal reference gene, and the 2^−ΔΔCt^ method was employed to calculate the relative expression of genes. Average values were visualized using Prism 10 (v10.1.2). Primer design software Oligo.7(v7) was used to design specific primers for the detected DEGs, and primer information is provided in Appendix A.

## 5. Conclusions

In this study, the transcriptomic profile of yak skin across different ages (0.5 years, 2.5 years, and 4.5 years) and body sites (scapular and ventral parts) was investigated. The Y2.5_S vs. Y2.5_V group had the most differentially expressed genes (491 DEGs), and the Y2.5_V vs. Y0.5_V comparison group had 370 DEGs across ages. Functional enrichment analysis indicated distinct immune differences by age and endocrine differences by body site. WGCNA found that immunity-related genes such as *GLYATL2*, *ACSL5*, *SPDEF*, and *FOXA1* were core in the scapula region module, likely due to more UV exposure. In the ventral region, genes related to hair follicle development and keratin (e.g., *FOXN1*, *OVOL1*, and *KRT27*) were hub genes, consistent with thicker ventral hair.

These results offer biological insights into yak skin biological function and adaptation. For breeding, they can guide efforts to improve wool quality and enhance yak health by targeting relevant genes. Future research should focus on validating candidate genes’ functions through experiments, integrating epigenetic data to understand gene regulation better, and linking gene expression to skin phenotypes. This will deepen our knowledge of yak skin biology and support sustainable yak resource use.

## Figures and Tables

**Figure 1 ijms-26-04601-f001:**
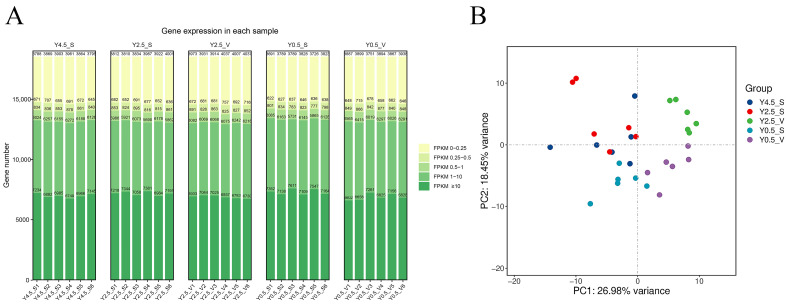
Gene expression analysis and principal component analysis. (**A**) The distribution of FPKM expression in each sample is represented by a stacked bar chart. (**B**) Results of group principal component analysis.

**Figure 2 ijms-26-04601-f002:**
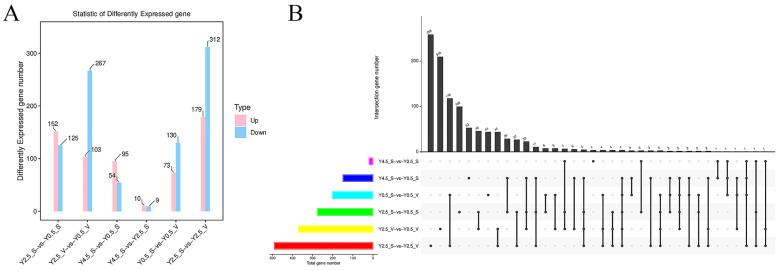
Statistics of differently expressed genes. (**A**) The quantity of DEGs between each group. (**B**) Upset diagram of DEGs.

**Figure 3 ijms-26-04601-f003:**
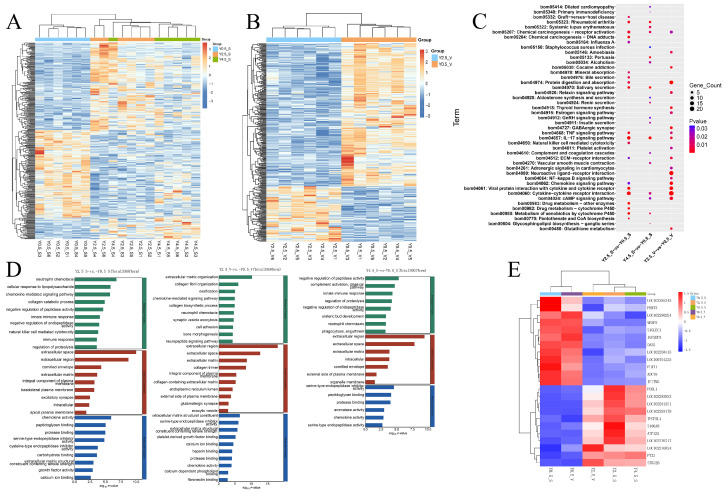
Expression patterns and functional analysis of DEGs at different ages. (**A**) Cluster heat map of the union of DEGs in the three groups (Y2.5_S vs. Y0.5_S, Y4.5_S vs. Y0.5_S, and Y4.5_S vs. Y2.5_S) of scapular skin of different ages. (**B**) Clustering heat map of DEGs in ventral skin of different ages (Y2.5_V vs. Y0.5_V). (**C**) Top 20 enriched KEGG pathways in comparison groups of different ages. (**D**) Top 30 significantly enriched GO terms, and all significantly enriched GO terms were displayed if values were less than 30. (**E**) Cluster heat map of the genes that were differentially expressed in all the three comparison groups.

**Figure 4 ijms-26-04601-f004:**
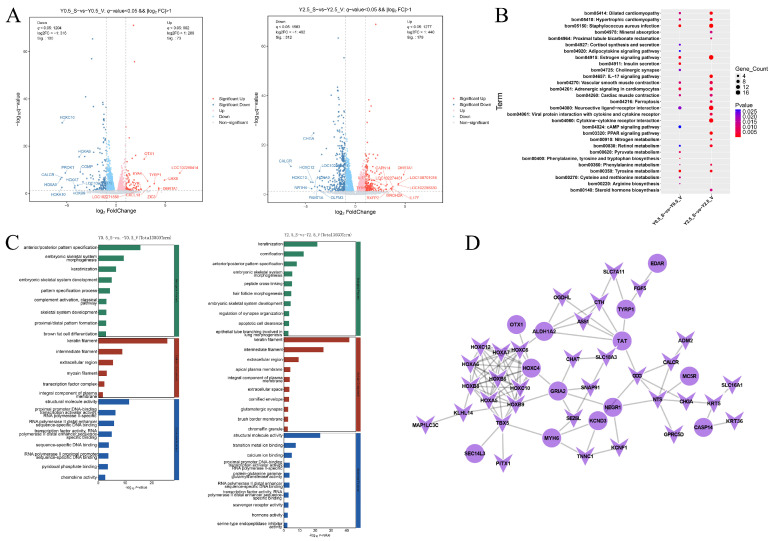
Differentially expressed genes at different regions and the functional enrichment analysis. (**A**) The volcano map of DEGs in different regions (Y0.5_S vs. Y0.5_V and Y2.5_S vs. Y2.5_V). (**B**) Top 20 enriched KEGG pathways in comparison groups of different regions. (**C**) Top 30 significantly enriched GO terms, and all significantly enriched GO terms were displayed if values were less than 30. (**D**) The PPI network of the DEGs that were commonly differentially expressed in the Y0.5_S vs. Y0.5_V and Y2.5_S vs. Y2.5_V groups; the V shape indicates downregulated, and circle indicates upregulated in the scapular region.

**Figure 5 ijms-26-04601-f005:**
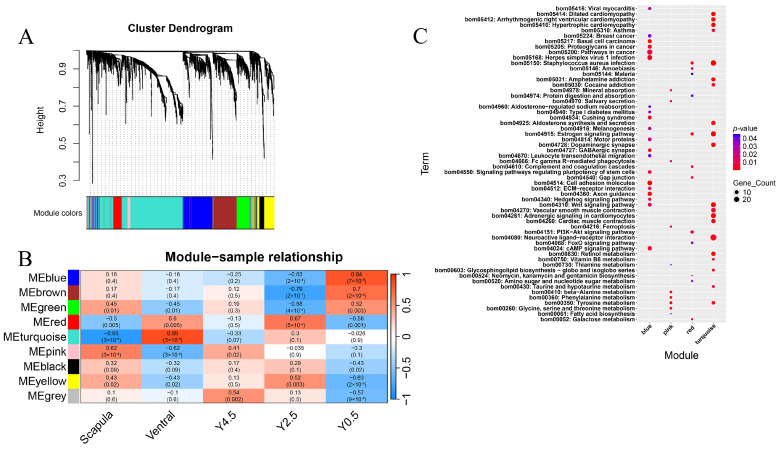
Weighted gene co-expression network analysis of DEGs (P adjust < 0.05 and FC ≥ 1.5) in different ages and different regions. (**A**) Cluster dendrogram constructed by WGCNA showing eight co-expressed gene modules; the co-expression modules are depicted in different colors, while the gray module indicates no correlation between genes. (**B**) Module–sample relationship of eight modules; the numbers above the heat map indicate the Person correlation coefficient (r) values. (**C**) Top 20 significantly enriched KEGG pathways of the key modules, and all significantly enriched KEGG pathways were displayed if values were less than 20.

**Figure 6 ijms-26-04601-f006:**
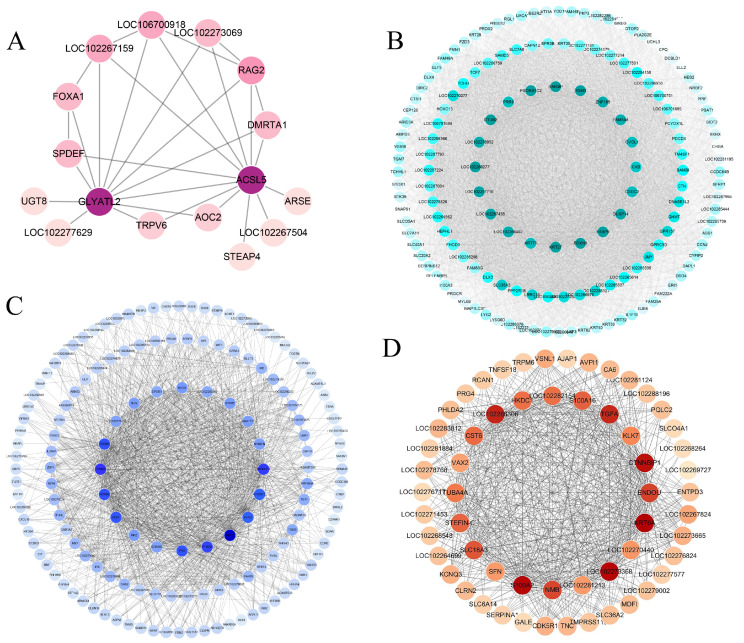
The co-expression network of the genes in the four key modules. (**A**) Co-expression network of the pink module with a significant positive correlation with the scapula. (**B**) Co-expression network of the turquoise module with a significant positive correlation with the ventral region. (**C**) Co-expression network of the blue module with a significant positive correlation with 0.5 years old. (**D**) Co-expression network of the red module with a significant positive correlation with 2.5 years old. All the networks were produced by the WGCNA package and visualized using the Cytoscape software. The hub genes of the top 150 degrees of connectivity in the turquoise and blue modules networks were calculated, and the genes in the pink and red module co-expression network were all exhibited. In each interaction network, the darker the color is, it indicates that the connectivity degree is higher.

**Figure 7 ijms-26-04601-f007:**
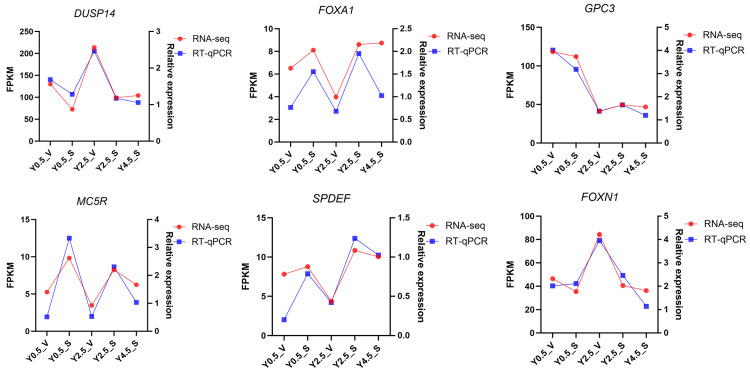
RT-qPCR and RNA-seq. RT-qPCR validation results for six randomly selected DEGs. Data points represent the mean value of three replicates.

## Data Availability

The data presented in the study are deposited in the NCBI repository, https://www.ncbi.nlm.nih.gov/sra/PRJNA1227113, (accessed on 23 April 2025), accession number PRJNA1227113.

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
