# Peer review of "Deciphering the Transcriptomic Complexity of Yak Skin Across Different Ages and Body Sites"

_ijms, 2025, doi:10.3390/ijms26104601_

Round 1
Reviewer 1 Report
Comments and Suggestions for Authors The manuscript presented by Zhang et al. shows the results of a study conducted on yaks,analyzing skin transcriptomics at different ages and taking into account two areas with
different UV light exposure. The work is particularly interesting due to the species'
extreme habitat; this could allow for a comparative analysis with the results found in
Bos taurus, since both species belong to the same genus. The introduction is well written,
but it would be necessary to add data from related species in which skin transcriptomic
studies were performed.
The methodology is adequate and well described. Regarding the molecular study, however,
some aspects of the choice of animals are confusing; especially considering the
importance of sex hormones on the skin, it would be necessary to determine whether o
nly males and females were used, and whether the 2.5-year-old animals had reached
sexual maturity. The transcriptomics results are well grouped for analysis. The results are understandable
because the graphics are good. In line with what was stated in the introduction, I believe the discussion should be
rewritten to consider comparative aspects. The analysis is intraspecific, considering
changes between skin regions and between ages, but the work would be greatly enriched
if comparative aspects and phylogenetic interests were incorporated into the
discussion. Furthermore, the authors emphasize the comparison with humans during aging, but the
oldest animals they use are 4.5 years old; is it valid to consider age-related
variations when the oldest animals are young adults?
Author Response
|
Comments 1: The manuscript presented by Zhang et al. shows the results of a study conducted on yaks, analyzing skin transcriptomics at different ages and taking into account two areas with different UV light exposure. The work is particularly interesting due to the species' extreme habitat; this could allow for a comparative analysis with the results found in Bos taurus, since both species belong to the same genus. The introduction is well written, but it would be necessary to add data from related species in which skin studies were performed. |
|
Response 1: Thank you very much for your insightful comments. Your suggestions are indeed valid and highly valuable to the improvement of our manuscript. However, as yaks are the only cattle breed capable of producing wool, and compared with yaks, the hair growth patterns of Bos taurus have not been a focus of research. As a result, we were unable to find relevant studies on the skin transcriptomics of Bos taurus for a direct comparison. In accordance with your suggestions, we have added research data on skin transcriptomic of other wool-producing species to the introduction section. The updated information can be found in lines 97 to 102 of the revised manuscript. |
|
Comments 2: The methodology is adequate and well described. Regarding the molecular study, however, some aspects of the choice of animals are confusing; especially considering the importance of sex hormones on the skin, it would be necessary to determine whether only males and females were used, and whether the 2.5-year-old animals had reached sexual maturity. |
|
Response 2: Thank you very much for your comments. In this study, all yaks were male yaks. The gender information has been added to the methods section, in line 439 of the revised manuscript. Moreover, it has been stated that the yaks at the age of 2.5 years were in the initial stage of sexual maturity. The relevant information has been supplemented in lines 299-300 of the discussion section in the revised manuscript. Comments 3: The transcriptomics results are well grouped for analysis. The results are understandable because the graphics are good. In line with what was stated in the introduction, I believe the discussion should be rewritten to consider comparative aspects. The analysis is intraspecific, considering changes between skin regions and between ages, but the work would be greatly enriched if comparative aspects and phylogenetic interests were incorporated into the discussion. Furthermore, the authors emphasize the comparison with humans during aging, but the oldest animals they use are 4.5 years old; is it valid to consider age-related variations when the oldest animals are young adults? Response 3: Thank you for your valuable comments. In the revised manuscript, most of the discussions section have been rewritten. A comparative study with the previous skin transcriptome research on the plateau adaptability of Tibetan goats has been added, which can be found in lines 406- 409; a comparative study on the gene expression of skin tissues from different parts of pigs has also been added, which can be found in lines 414-417. Furthermore, although yaks at the age of 4.5 years old are in their middle- age stage, the skin phenotype of these yaks, especially that of the dorsal skin, shows the characteristics of being prone to hair loss and pigment deposition. Therefore, 4.5 years old yaks as the oldest animals in this study has a certain validity. The relevant described has been added to lines 341-343 of the discussion section in the revised manuscript. |
Reviewer 2 Report
Comments and Suggestions for Authors
The manuscript is scientifically sound, the experimental design is comprehensive, involving different age groups and body sites, and the results are well presented. However, there are several areas that need improvement to enhance the quality of the manuscript.
1) There are a few minor grammar and punctuation errors.
Line 15 - The expression "in different ages" is inaccurate and can be changed to "at different ages".
Line 38 - The use of "and" here is incorrect.
Line 45 - "Skin is composed of three main layers of epidermis, dermis and hypodermis." It should be "Skin is composed of three main layers: epidermis, dermis, and hypodermis."
Lines 52-53 “change” instead of “changes”
Line 71 - "obviously observed" is not grammatically correct.
2) More animal information details about the yaks used in the study are required. Information such as the breed of the yaks, forms of livestock rearing, and any potential genetic variations within the population could affect the results. This information should be provided to ensure the reproducibility of the study.
3) How many biological and technical replicates have been used in qPCR?
Author Response
|
Comments 1: The manuscript is scientifically sound, the experimental design is comprehensive, involving different age groups and body sites, and the results are well presented. However, there are several areas that need improvement to enhance the quality of the manuscript. 1) There are a few minor grammar and punctuation errors. Line 15 - The expression "in different ages" is inaccurate and can be changed to "at different ages". Line 38 - The use of "and" here is incorrect. Line 45 - "Skin is composed of three main layers of epidermis, dermis and hypodermis." It should be "Skin is composed of three main layers: epidermis, dermis, and hypodermis." Lines 52-53 “change” instead of “changes” Line 71 - "obviously observed" is not grammatically correct. |
|
Response 1: Thank you for pointing this out. We apologize for this grammatical problem and have corrected it based on your suggestions. The changes marked in red are the revised portion according to the specific comments. Line 15- The “in different ages” was revised “at different ages” in the revised manuscript. It is on line 17 in the revised manuscript. Line 38- We added "They" after "and" to make the sentence structure clearer in line 43 of the revised manuscript. Line 45- The sentence in line 45 was changed to "Skin is composed of three main layers: epidermis, dermis, and hypodermis." Lines 52-53 The “change” was revised “changes” in line 59 of the revised manuscript. Line 71 – The sentence was changed to “It was observed that the back hair of older yaks was obviously more prone to shedding” in line 79 of the revised manuscript. |
|
Comments 2: More animal information details about the yaks used in the study are required. Information such as the breed of the yaks, forms of livestock rearing, and any potential genetic variations within the population could affect the results. This information should be provided to ensure the reproducibility of the study. |
|
Response 2: Detailed information about animal husbandry, etc. has been added to the Methods section in lines 439-442 of the revised manuscript. Comments 3: How many biological and technical replicates have been used in qPCR? Response 3: Three yaks were randomly selected from each group for qPCR detection. So, there were three biological replicates and three technical replicates in qPCR. |
Reviewer 3 Report
Comments and Suggestions for Authors
General Comments
This manuscript explores the transcriptomic landscape of yak skin across different ages and anatomical sites using WGCNA and differential gene expression analysis. The study contributes to the understanding of yak skin biology and its adaptation to the extreme plateau environment. The paper is generally well-organized and provides valuable datasets; however, several areas need improvement before publication.
Specific Comments
Introduction
1. The Introduction provides background on yaks and their environment but fails to define a precise research gap. What exact knowledge is lacking about yak skin development or transcriptomics that this study aims to fill?
2. The motivation seems too broad. For example, mentioning yak adaptation is fine, but it is unclear how transcriptomic complexity in skin contributes to adaptation or why different body parts matter at the transcriptomic level.
3. A clear hypothesis is missing. The objectives are not clearly stated. Is the aim to find age-specific, region-specific genes or understand the underlying mechanisms of adaptation?
4. It should briefly explain why WGCNA is suitable for this type of study and what is known about gene expression in skin development or anatomical regions.
Materials and Methods
1. The manuscript does not provide clear sample numbers per group (age and body site), replication, or randomization strategy.
2. There is no mention of RNA integrity number (RIN), library construction kits used, sequencing depth (in millions of reads/samples), or mapping efficiency.
3. The power selection process, criteria for module detection, and parameters used in WGCNA are not explained. It’s unclear how the soft-thresholding power was chosen.
4. It is not clearly stated which databases were used (e.g., KEGG, GO), and multiple test correction (FDR) is not mentioned.
Results
1. It would improve readability and impact if authors cross-linked DEGs and WGCNA modules with shared pathways or co-regulated networks.
2. Most of the text in the figures is difficult to read. Please enhance the visibility and clarity of all figures.
Discussion
1. The Discussion largely repeats results without enough insight into biological implications. For instance, how do DEGs relate to the physiological or adaptive traits of yak skin?
2. Although several genes/modules are discussed, little connection is made to known skin functions (e.g., barrier function, thermoregulation, immune defense) or adaptation strategies in high-altitude mammals.
3. There is no comparison with other species or previous transcriptomic studies in cattle or other plateau animals to contextualize the findings.
4. The manuscript lacks a limitations paragraph, such as the possible confounding effect of environmental variables, individual variation, or lack of proteomic/phenotypic validation.
Conclusion
1. The conclusion summarizes findings but does not extrapolate to biological or practical implications, such as insights into yak breeding, health, or skin applications.
2. It misses out on future research directions (e.g., functional validation of candidate genes, integration with epigenetic data, or phenotypic correlations).
Author Response
|
Comments 1: This manuscript explores the transcriptomic landscape of yak skin across different ages and anatomical sites using WGCNA and differential gene expression analysis. The study contributes to the understanding of yak skin biology and its adaptation to the extreme plateau environment. The paper is generally well-organized and provides valuable datasets; however, several areas need improvement before publication. Specific Comments Introduction 1. The Introduction provides background on yaks and their environment but fails to define a precise research gap. What exact knowledge is lacking about yak skin development or transcriptomics that this study aims to fill? 2. The motivation seems too broad. For example, mentioning yak adaptation is fine, but it is unclear how transcriptomic complexity in skin contributes to adaptation or why different body parts matter at the transcriptomic level. 3. A clear hypothesis is missing. The objectives are not clearly stated. Is the aim to find age-specific, region-specific genes or understand the underlying mechanisms of adaptation? 4. It should briefly explain why WGCNA is suitable for this type of study and what is known about gene expression in skin development or anatomical regions.
|
|
Response 1: 1. Thank you for pointing this out. This study aims to address these gaps by analyzing the transcriptomes of yak skin from different ages and body parts, identifying key genes and gene networks, and uncovering the molecular changes of yak skin biology. We added the description regarding the research gap in lines 81-82 and lines 102-103 in the introduction section of the revised manuscript. 2. Thank you for your comment. Yaks live in harsh high - altitude environments, different skin parts face varying environmental pressures and have distinct functions. For example, the back is more exposed to UV and has seasonally - shedding downy hairs, while the abdomen has long, coarse hairs for protection. Transcriptomic differences among parts may help yaks adapt, and our study analyzes these to uncover the molecular adaptation mechanisms, which description could be found in lines 77-92. 3. Our hypothesis is that yak skin has age - and region - specific gene expression patterns linked to development and adaptation, with genes interacting in co - expression networks. Our objectives are to identify such differentially expressed genes and understand the adaptation mechanisms via gene co - expression network analysis and functional studies. The hypothesis was added in lines 104-107 of the revised manuscript. 4. WGCNA is ideal for our study as it can cluster genes with similar expression patterns, helping to explore gene networks in complex yak skin transcriptomic data. It was briefly explained in lines 108-111 of revised manuscript. Currently, research on gene expression in different anatomical regions of animal skin is limited. The studies on mammalian skin transcriptomics primarily focus on the research of hair follicle cycle development in wool-producing animals or comparative studies on wool fineness. We added the description in lines 97-102 of the revised manuscript. |
|
Comments 2: Materials and Methods 1. The manuscript does not provide clear sample numbers per group (age and body site), replication, or randomization strategy. 2. There is no mention of RNA integrity number (RIN), library construction kits used, sequencing depth (in millions of reads/samples), or mapping efficiency. 3. The power selection process, criteria for module detection, and parameters used in WGCNA are not explained. It’s unclear how the soft-thresholding power was chosen. 4. It is not clearly stated which databases were used (e.g., KEGG, GO), and multiple test correction (FDR) is not mentioned.
|
|
Response 2: Thank you for your comment. I have carefully revised the Materials and Methods. 1. Sample numbers per group (6 yaks per age and body site), six replicates for each sample, and the random selection of samples have been clearly presented in the lines 439-441 of the revised manuscript. 2. The RNA Integrity Numbers (RIN) and library construction kits were described in lines 454-456, that is “(RINs) ≥ 6.5 and 28S/18S ratios ≥ 1.0, were used to construct the sequencing library. Then the libraries were constructed using VAHTS Universal V6 RNA-seq Library Prep Kit according to the manufacturer’s instructions.” Sequencing depth (in millions of reads/samples), or mapping efficiency was described in lines 122-125 of the result section, that is “After quality control, high-quality clean reads were obtained, with an effective data volume of 6.86-7.05 G per sample, and the Q30 values ranged from 95.74% to 96.45%, with an average GC content of 50.52%. The mapping rate to the yak reference genome using HISAT2 ranged from 93.27% to 96.5% (Table S1)”. 3. The power selection process was added and supported by Figure S2, and the parameters used in WGCNA are explained in lines 482-486 of the revised manuscript. 4. Regarding the databases used in our study, we primarily utilized the KEGG (Kyoto Encyclopedia of Genes and Genomes) and GO (Gene Ontology) databases. In the revised manuscript, we have added a more detailed description in the Materials and Methods section (lines 471 - 473) to clarify the use of these databases. The False Discovery Rate (FDR) information was added in lines 467-468 of the revised manuscript. Comments 3: Results 1. It would improve readability and impact if authors cross-linked DEGs and WGCNA modules with shared pathways or co-regulated networks. 2. Most of the text in the figures is difficult to read. Please enhance the visibility and clarity of all figures. Response 3: 1. 1. Thank you for your insightful comments. In the revised manuscript, we have cross-linked and described the shared pathways between DEGs and WGCNA modules in the results section of the WGCNA KEGG analysis (lines 246- 254). 2. 2. In the revised manuscript, the resolution of all figures has been increased to 300 or 600 dpi to enhance the visibility and clarity of the figures. |
Comments 4: Discussion
1. The Discussion largely repeats results without enough insight into biological implications. For instance, how do DEGs relate to the physiological or adaptive traits of yak skin?
2. Although several genes/modules are discussed, little connection is made to known skin functions (e.g., barrier function, thermoregulation, immune defense) or adaptation strategies in high-altitude mammals.
3. There is no comparison with other species or previous transcriptomic studies in cattle or other plateau animals to contextualize the findings.
4. The manuscript lacks a limitations paragraph, such as the possible confounding effect of environmental variables, individual variation, or lack of proteomic/phenotypic validation.
Response 4:
1. Thank you for your comments. I have deleted part of the discussion that repeated the results. Additionally, I have added content regarding the biological functions of DEGs and their roles in the adaptability of yak skin. This added content could be found in lines 366-373 of the revised manuscript.
2. In this study, since we analyzed the differentially expressed genes only among the skins of the same species, the enriched functions did not obviously reflect the skin barrier function or the high-altitude adaptation strategies. However, the differential analyses of different ages and different body parts both enriched the functions related to immunity, inflammation, and infection that were enriched also reflect the known functions of the skin. The relevant descriptions are involved in the introduction.
3. Since yaks are rare cattle species that produce cashmere, there are very few studies on the skin tissues of common cattle. In the discussion section of the revised manuscript, a comparative study with the previous skin transcriptome research on the plateau adaptability of Tibetan goats has been added, which can be found in lines 406- 409; a comparative study on the gene expression of skin tissues from different parts of pigs has also been added, which can be found in lines 415-419.
4. Thank you for your valuable comments. A paragraph on study limitations has been added based on the research content and results, which can be found in lines 426-434 of the revised manuscript.
Comments 5: Conclusion
1. The conclusion summarizes findings but does not extrapolate to biological or practical implications, such as insights into yak breeding, health, or skin applications.
2. It misses out on future research directions (e.g., functional validation of candidate genes, integration with epigenetic data, or phenotypic correlations).
Response 5:
1. The conclusion was revised. Added content on biological implications and practical applications in lines 528-529 in the revised manuscript.
2. Thank you for your comments. The future research directions have been added at the end of the discussion section, in lines 530- 533 of the revised manuscript.
Round 2
Reviewer 1 Report
Comments and Suggestions for Authors
The manuscript can be accepted. Only must be corrected the introduction because hypodermis is not a skin layer (is part of the tegument, but skin layers are Epidermis and Dermis)
Author Response
Comments: The manuscript can be accepted. Only must be corrected the introduction because hypodermis is not a skin layer (is part of the tegument, but skin layers are Epidermis and Dermis)
Response: Thank you for your valuable comments. The sentence in the introduction that you mentioned has been revised: "The skin is primarily composed of two layers: epidermis and dermis.” (on line 50).
Once again, thank you for your support and assistance. Best regards, Xiaolan ZhangReviewer 3 Report
Comments and Suggestions for Authors
No further comments.
Author Response
Comments : No further comments.
Response: Thank you for your review and acceptance. We appreciate your support and assistance.